# Pregnancy, Breastfeeding, and Vitamin D

**DOI:** 10.3390/ijms241511881

**Published:** 2023-07-25

**Authors:** Teodoro Durá-Travé, Fidel Gallinas-Victoriano

**Affiliations:** 1Department of Pediatrics, School of Medicine, University of Navarra, 31008 Pamplona, Spain; 2Navarrabiomed (Biomedical Research Center), 31008 Pamplona, Spain; 3Department of Pediatrics, Navarra Hospital Complex, 31008 Pamplona, Spain; fivictoriano@hotmail.com

**Keywords:** pregnancy, breastfeeding, human breast milk, breastfed infants, metabolism, supplementation, vitamin D

## Abstract

Exclusive breastfeeding is considered the ideal food in the first six months of life; however, paradoxically, vitamin D content in human breast milk is clearly low and insufficient to obtain the recommended intake of 400 IU daily. This article summarizes the extraordinary metabolism of vitamin D during pregnancy and its content in human breast milk. The prevalence of hypovitaminosis D in pregnant women and/or nursing mothers and its potential maternal–fetal consequences are analyzed. The current guidelines for vitamin D supplementation in pregnant women, nursing mothers, and infants to prevent hypovitaminosis D in breastfed infants are detailed. Low vitamin D content in human breast milk is probably related to active changes in human lifestyle habits (reduced sunlight exposure).

## 1. Introduction

Exclusive breastfeeding, owing to its specific composition that adapts to the needs of infants, is considered the ideal nourishment during the first six months of age in order to achieve optimal health, growth, and development. From that moment forward, with the intention to fulfill the nutritional requirements, adequate complementary feeding should be gradually introduced without breastfeeding withdrawal even beyond two years of age [1]. However, it may sound paradoxical that breastfed babies require vitamin D pharmacological supplements.

Vitamin D metabolism manifests significant changes in pregnant women in comparison to the non-pregnant state [2,3,4,5], but several questions about the role of vitamin D in pregnancy remain unanswered. Vitamin D deficiency has been reported among pregnant women and nursing mothers globally, constituting a risk group for vitamin D deficiency [2,6,7,8]. Vitamin D deficiency during pregnancy has been associated not only with pregnancy outcomes but also with the posterior physical and mental health of the offspring [9,10,11,12].

Conventionally, vitamin D’s main physiological function was thought to be the maintenance of calcium and phosphorous serum concentrations in a physiological range. To do so, 1,25(OH)2D improves dietary calcium and phosphate absorption, increases calcium tubular reabsorption, and modifies calcium and phosphate bone deposition and resorption. However, at present, vitamin D is considered a pleiotropic hormone (its biological action is performed in different organ systems). In fact, in addition to the recognized role in calcium homeostasis and bone metabolism (calcium deficiency causes rickets in infants and osteomalacia in adults), vitamin D regulates the expression of a variety of genes that control multiple metabolic pathways related to immune function, as well as proliferation, differentiation and cell metabolism [2,13]. This information helps explain the fundamentals of a large number of recent publications that associate vitamin D deficiency with a higher risk of several chronic diseases (autoimmune, cardiovascular, infectious, metabolic, neurologic, psychiatric diseases, and several types of cancer) and, therefore, justify the concern on the monitoring of its organic content.

The main objective of this work is to accomplish a review of vitamin D metabolism during pregnancy and breastfeeding and the content in breast milk by means of the analysis of the prevalence of hypovitaminosis and the potential maternal–fetal consequences. We also detail current trends in vitamin D supplementation for mothers during pregnancy and breastfeeding and for infants in order to prevent hypovitaminosis D.

## 2. Vitamin D Metabolism

In order to appreciate the uniqueness of vitamin D metabolism during gestation, a brief physiological review of human vitamin D metabolism under normal circumstances is essential [2,13,14,15].

The concept of vitamin D does not refer exclusively to a single compound but rather encompasses different chemical structures with complementary biological functions. Approximately 80–90% of the body’s vitamin D is of endogenous origin (cholecalciferol or vitamin D3) and is synthesized from skin exposure to sun radiation; meanwhile, a much smaller amount has a dietary origin, either as cholecalciferol from animal products or as ergocalciferol, or vitamin D2 from plants, fungi, or yeast photosynthesis. Both compounds are biologically inactive and require a double (hepatic and renal) hydroxylation for their functional activation. The exposure to type B ultraviolet radiation induces endogenous synthesis of cholecalciferol from epidermal dehydrocholesterol (provitamin D3), turning into precholecalciferol (previtamin D3), a compound that is thermodynamically unstable and rapidly transforms into cholecalciferol under the influence of body temperature.

Both animal (cholecalciferol) and plant (ergocalciferol) origin vitamin D compounds and those derived from cutaneous synthesis after the photolysis of 7-dehydrocholesterol (cholecalciferol) reach the bloodstream and are transported to the liver bound to a specific protein, vitamin D binding protein (VDBP). Within the hepatocytes, the enzyme cholecalciferol-25-hydroxylase (also known as CYP2R1) catalyzes its first hydroxylation, giving rise to 25-hydroxycholecalciferol [25(OH)D] or calcidiol. The 25(OH)D released into the blood circulation is transported bound to VDBP to the kidneys, where the enzyme 1-α-hydroxylase (also known as CYP27B1), mainly in the renal tubule, catalyzes a second hydroxylation that converts 25(OH)D to 1,25-hydroxycholecalciferol [1,25(OH)2D] or calcitriol, the compound that constitutes the biologically active form of vitamin D. The biological functions of vitamin D are mediated by a nuclear transcription factor known as the vitamin D receptor (VDR). The binding of 1,25(OH)2D to its VDR initiates a cascade of molecular interactions that modulate the transcription of different genes involved not only in the regulation of mineral and bone metabolism but also in a large variety of biological functions.

The main regulators of 1,25(OH)2D serum levels are parathyroid hormone (PTH), serum calcium and phosphate concentration, and 1,25(OH)2D itself. In fact, 1,25(OH)2D regulates its own catabolism by feedback mechanisms in such a way that its high concentrations get to activate the functionality of the CYP24A1 enzyme (24-hydroxylase activity) that catalyzes the conversion of 25(OH)D and 1,25(OH)2D into 24,25(OH)2D and 1,24,25(OH)3D, respectively, (vitamin D metabolites with practically no biological activity) and, simultaneously, it reduces the enzymatic activity of CYP27B1, resulting in a decrease in 1,25(OH)2D levels.

The majority of cells and organs (placenta, lung, muscle, heart, bone marrow, blood vessels, brain, breast, colon, prostate, parathyroid glands, pancreas, thyroid, gonads, skin, fat tissue, activated B- and T-lymphocytes, etc.) contain RVD and activating enzymes (1-alpha-hydroxylase) for synthesizing 1,25(OH)2D, probably enabling exertion of a local autocrine or paracrine function. This circumstance raises the question of the biological importance that sufficient organic levels of vitamin D would presumably play.

## 3. Definition of Hypovitaminosis D

Growing evidence that vitamin D deficiency may imply serious health consequences beyond rickets and/or osteomalacia justifies the need for an accurate assessment of body vitamin D status.

The determination of serum 25(OH)D concentration (1 ng/mL corresponds to 2.5 nmol/L) is currently considered to be the best indicator to assess the body status of vitamin D. The half-life of 25(OH)D is 2 to 3 weeks, much longer than that of the active metabolite 1,25(OH)2D, whose half-life is only 4 h. Furthermore, it should be noted that 1,25(OH)2D is not a good indicator of total vitamin D content because slight decreases in serum calcium concentration lead to an increase in PTH that could induce a boost in its renal synthesis, resulting in normal or even elevated 1,25(OH)2D values [13,14].

Even though it is generally accepted that serum 25(OH)D levels are the best indicator of body vitamin D stores, there is some controversy with regard to the limits that define normality. In 2011, the American Institute of Medicine (IOM) proposed considering 25(OH)D levels of 20 ng/mL as the threshold of normality for the organic vitamin D content, given the practical absence of significant metabolic alterations at concentrations above 20 ng/mL, while levels below these figures induce an increase in PTH and bone alterations might be already detected [16]. However, in 2012, the US Endocrine Society considered it prudential, in order to safeguard the multiple extra-skeletal functions of vitamin D, to modify the 25(OH)D values defining vitamin D sufficiency without modifying the already established cut-off point for vitamin D deficiency but intercalating between these two states the concept of vitamin D insufficiency, which should also be overcome [17,18].

These new cut-off points, although based on observational studies, are being accepted and used by most authors:-Vitamin D deficiency: calcidiol levels are lower than 20 ng/mL (<50 nmol/L);-Vitamin D insufficiency: calcidiol levels range between 20 and 29 ng/mL (51–74 nmol/L);-Vitamin D sufficiency: calcidiol levels are equal to or higher than 30 ng/mL (>75 nmol/L).

## 4. Vitamin D Metabolism during Pregnancy

Vitamin D metabolism during pregnancy is unique, to the point that some authors have described it as extraordinary [3] since it differs significantly from the preconceptional maternal condition or that of the non-pregnant woman. Moreover, it becomes more notorious since during gestation, the mother is the exclusive source of vitamin D essential for fetal development, and, in fact, if the pregnant mother is deficient in vitamin D, the fetus and/or newborn will also be deficient [19]. The most significant changes in vitamin D metabolism during gestation include (a) a progressive increase in maternal serum levels of 1,25(OH)2D, a product that does not cross the placental barrier, (b) an increase in maternal renal and placental CYP27B1 enzyme activity, (c) an increase in maternal VDBP levels, and finally, (d) a decrease in 1,25(OH)2D catabolism [4,20]. The placenta plays an important role in the metabolism and delivery of vitamin D to the fetus, as both its maternal (decidual) and fetal (trophoblastic) sides contain VDR, as well as the CYP27B1 enzyme that launches the conversion of 25(OH)D to 1,25(OH)2D, which is the biologically active form of vitamin D [5,15].

There is a significant increase in the maternal renal conversion of 25(OH)D to 1,25(OH)2D during pregnancy. In this process, the maternal side of the placenta contributes considerably as an extrarenal organ in the activation of vitamin D [21]. In this way, at the end of the first trimester of gestation, serum levels of 1,25(OH)2D practically triple in comparison to the preconceptional stage without being associated with changes in serum calcium and PTH levels, reaching levels of 1,25(OH)2D that are hypothetically “toxic” due to the condition of hypercalcemia that they are supposed to entail [19]. The increase in maternal 1,25(OH)2D levels would foster the intestinal absorption of calcium and, consequently, a secondary increase in its plasma levels in order to cover, through its transplacental passage, the fetal calcium demands for its skeletal mineralization [22]. In other words, during gestation, a “physiological mismatch” in the interaction of vitamin D and serum calcium levels seems to manifest itself; in fact, the maternal increase in 1,25(OH)2D levels depends exclusively on the availability of 25(OH)D, being independent of calcium homeostasis [19].

It is interesting to note the biphasic relationship between 25(OH)D and 1,25(OH)2D serum levels during gestation. Maternal 25(OH)D levels remain practically constant during pregnancy, showing a direct correlation between serum concentrations of both vitamin D metabolites, despite supraphysiological levels of 1,25(OH)2D. However, when maternal 25(OH)D levels exceed 40 ng/mL, maternal serum 1,25(OH)2D concentration, although at “supraphysiological” levels, also remains constant (Figure 1), suggesting that optimizing maternal renal/placental 1,25(OH)2D production during gestation requires circulating 25(OH)D levels of at least 40 ng/mL [19,23].

On the other hand, from the placental formation (at 4 weeks of gestation) to the end of pregnancy, fetal and/or umbilical cord serum 25(OH)D levels correlate with maternal 25(OH)D levels as a consequence of the transplacental passage of 25(OH)D (it was previously referred that 1,25(OH)2D does not cross the placental barrier) [6,9,24]. Thus, fetal 1,25(OH)2D levels will depend exclusively on the fetal renal and placental conversion of maternal 25(OH)D to 1,25(OH)2D by the CYP27B1 enzyme [25]. Maternal VDBP levels also increase significantly during gestation [12], and, given its higher affinity for 25(OH)D with respect to 1,25(OH)2D, it has been suggested that this increase in VDBP may play a role as a reservoir and/or redistribution of 25(OH)D into the fetal circulation.

As previously mentioned, 1,25(OH)2D would regulate its own catabolism through feedback mechanisms in non-pregnant women. This process implies that its high concentrations would increase the activity of the CYP24A enzyme and, therefore, an increase in 24,25(OH)2D and 1,24,25(OH)3D, as well as an inhibition of the activity of the CYP27B1 enzyme and consequently, a decrease in the levels of 1,25(OH)2D. However, during pregnancy, another “physiological mismatch” occurs due to methylation at the maternal–fetal placental level, which reduces the functionality of the CYP24A1 enzyme and, consequently, suppresses the activation of the vitamin D catabolism mechanism [2,26]. This other “mismatch” would contribute, to a large extent, to the increase in maternal levels of circulating 1,25(OH)2D during pregnancy, ensuring a higher bioavailability of 1,25(OH)2D and maternal–fetal calcium.

Calcitonin and parathyroid hormone-related peptide (PTHrP), in conjunction with other factors (IGF-1, placental lactogen, estradiol, and prolactin), seem to have a determining role in the metabolism of vitamin D during gestation, favoring the transplacental passage of calcium, as well as stimulating the activity of the CYP27B1 enzyme, which would explain, to a large extent, the significant increase in 1,25(OH)2D in pregnant women and the suppression of PTH levels during pregnancy [12]. PTHrP secretion takes place in the maternal mammary tissue, the placenta in its maternal and fetal sides, and in the fetal parathyroid glands, and, apart from its functions already described, it seems to have a maternal bone protective function (it inhibits osteoclastic activity) and contributes to fetal endochondral ossification [27].

In short, vitamin D metabolism during gestation presents peculiar characteristics, possibly of a teleological nature. This means, given that vitamin D plays a modulating role in the immune system, these apparent “physiological mismatches” would be rather epigenetic maternal–fetal adaptation mechanisms in order to maintain immunological homeostasis and, thus, contribute to maternal–fetal immunological tolerance rather than playing an exclusive endocrinological role in the regulation of calcium homeostasis [2,28,29].

## 5. Vitamin D Deficiency in Pregnant and Nursing Mothers

Vitamin D status is usually conditioned by various factors, such as skin pigmentation, physical agents that block exposure to solar radiation (clothing, sunscreens, etc.), and geographical variables (latitude, climate, season, altitude, etc.) [2,6]. Therefore, these factors should be considered before making comparisons between vitamin D deficiency prevalence figures from countries with different ethnic, cultural, and/or geographical characteristics. In fact, vitamin D deficiency in pregnant women and/or nursing mothers ranges from 24–77% in Western countries (Belgium, Canada, Germany, Spain, United States, Netherlands, United Kingdom) to 46–97% in Asian and African countries (China, India, Iran, Nigeria, Pakistan, Kenya, Kuwait, Turkey) [2,6,7,8].

In summary, regardless of ethnic, sociocultural, or geographical differences, vitamin D deficiency among pregnant women represents a health problem of considerable global dimensions, constituting a relative risk group for vitamin D deficiency. Therefore, it would be a priority to develop prevention strategies (sufficient sun exposure, intake of vitamin D-fortified foods, and pharmacological vitamin D supplementation) and generalized screening of pregnant women (serum 25(OH)D determination).

## 6. Consequences of Vitamin D Deficiency during Pregnancy

Vitamin D deficiency during gestation, in addition to potentially causing connatal or early postnatal rickets in the newborn [30], has been associated with repeated embryo implantation failure and recurrent fetal loss [31]. It has also been associated with an increased risk of gestational hypertension and preeclampsia, gestational diabetes, and cesarean delivery, as well as prematurity and/or intrauterine growth retardation [9,10,32]. It has also been suggested that maternal–fetal vitamin D deficiency could condition various long-term pathologies through phenomena of “fetal programming” and/or epigenetic modification, such as mineralization defects, alterations in body composition, bronchial asthma, atopic and/or autoimmune diseases, and neurodevelopmental diseases (psychomotor and/or language delay, attention deficit hyperactivity disorder, autism, etc.) [2,11,12,30,33].

## 7. Maternal Vitamin D Supplementation during Pregnancy

There is great controversy among professionals as to whether or not to institute generalized pharmacological vitamin D supplementation during pregnancy; moreover, there is not even a clear consensus on the optimal time period and dose of vitamin D to be administered among those countries that have decided to recommend selective and/or generalized vitamin D supplementation for pregnant women.

The authors, who question generalized pharmacological supplementation with vitamin D and advocate for more moderate measures, such as regulated sun exposure and intake of vitamin D-fortified foods in combination with analytical screening during pregnancy and pharmacological supplementation with vitamin D exclusively in pregnant women with hypovitaminosis D (deficiency and insufficiency), argue that the changes in vitamin D metabolism that take place during pregnancy are still largely unexplained. For example, circulating levels of 1,25(OH)2D during pregnancy reach “supraphysiological” values, even up to 300 pg/mL, whereas in other conditions (non-pregnant women), levels of only 80 pg/mL could be accompanied by severe hypercalcemia [19].

Fetal and/or neonatal serum 25(OH)D levels are directly related to maternal 25(OH)D levels. In this way, vitamin D deficiency in the pregnant woman would result in a lower placental transfer of 25(OH)D and, consequently, a lower accumulation or deposit of vitamin D in the newborn [13,23,34,35]. Given the high prevalence of vitamin D deficiency in pregnant women, as well as the importance of its potential consequences on maternal–fetal health, it seems risky to rely exclusively on sun exposure and the frequent intake of vitamin D-fortified foods as priority measures to achieve sufficient vitamin D status. Therefore, pharmacological supplementation of vitamin D (cholecalciferol) during pregnancy would be recommended [36,37]. In fact, a recent meta-analysis published by the Cochrane Library corroborates that vitamin D supplementation during gestation would reduce the maternal risk of preeclampsia and/or gestational diabetes, as well as intrauterine growth retardation and prematurity [10].

There is currently no uniform criterion establishing the dose of pharmacological vitamin D supplementation in pregnancy required to achieve maternal circulating levels of 25(OH)D of at least 40 ng/mL in order to optimize renal/placental production of maternal 1,25(OH)2D during gestation [19,23,38]. For example, while the IOM suggests an oral dose of 600 IU daily [39], the US Endocrine Society recommends a daily oral dose of 1500–2000 IU [17]. However, randomized controlled clinical studies have shown that daily supplementation with 600 IU of vitamin D during gestation is inadequate to achieve sufficient maternal circulating levels of 25(OH)DE, whereas maternal vitamin D supplementation with 2000 or 4000 IU daily achieves maternal 25(OH)D levels above 30 and 40 ng/mL, respectively, without any adverse maternal–fetal effects (hypercalcemia, hypercalciuria, hypocalcemia, and hypervitaminosis D) [36,40,41,42]. Recent observational studies have even emphasized the importance of maintaining sufficient maternal vitamin D levels in the first trimester of pregnancy and/or preconception period in relation to the risk of repeated embryo implantation failure in situations of vitamin D deficiency. Although randomized controlled studies are required, these results suggest that pharmacological supplementation with vitamin D in pregnant women should begin as early as possible [43].

## 8. Vitamin D Deficiency in the Neonatal Period

Apart from the immediate maternal–fetal consequences of vitamin D deficiency during pregnancy that have been mentioned above, maternal vitamin D deficiency entails the risk of altered fetal and/or postnatal vitamin D status [23,34,35]. A recent meta-analysis involving pregnant women and their newborns from different population groups corresponding to the WHO regions from which data are available (Americas, Southeast Asia, Europe, Eastern Mediterranean, and Western Pacific) indicates that more than half of the mothers at the end of pregnancy and their newborns were vitamin D deficient (although with some variability among the different geographic areas included in this study). Furthermore, there was a significant correlation between maternal and newborn umbilical cord 25(OH)D levels, with maternal levels being logically significantly higher, given the impossibility of the fetus to synthesize 25(OH)D acquired from the mother via the transplacental route [7]. In other words, the prevalence of vitamin D deficiency in both pregnant women and their newborns is considerable, making it a priority to standardize prevention strategies that, in the case of the infants, protect them from the potential adverse effects in the short and/or long term, of vitamin D deficiency.

Obviously, vitamin D supplementation would be the simplest and safest measure to prevent not only childhood rickets and/or bone health but also, given its extra-skeletal functions, overall health status. Classically, vitamin D deficiency has been related to recurrent lower respiratory tract infections (pneumonia, bronchiolitis, etc.), a condition classically called “rickets lung”, but hypovitaminosis D has even come to be related to different types of infectious diseases (urinary tract infections, otitis media, acute diarrhea, sepsis, etc.), including COVID-19. Thus, it has been speculated whether exclusive breastfeeding and infant supplementation with vitamin D could have a positive synergistic effect in the prevention of infectious diseases in infancy [44,45,46].

## 9. Vitamin D Content in Breast Milk

The predominant forms of vitamin D in breast milk are cholecalciferol and 25(OH)D. The estimation of the vitamin D content or its so-called “anti-rachitic activity” is based on the activity and/or biological effects of its metabolites. For example, 40 IU of vitamin D would correspond to the biological equivalent of one microgram of cholecalciferol (1 μg = 40 IU), whereas the biological equivalent of one microgram of 25(OH)D would correspond to 200 IU of vitamin D (1 μg = 200 IU). The reason is that oral administration of 25OHD is five times more effective than cholecalciferol in raising circulating concentrations of 25OHD [35,47,48,49].

Vitamin D content in breast milk is quite stable, even in prolonged breastfeeding, with seasonal variations. Its content depends on the maternal vitamin D status and, consequently, increases with pharmacological vitamin D supplementation to lactating mothers [34,35,49]. Comparative studies carried out in lactating mothers of different ethnic, cultural, and geographical conditions, while confirming the existence of variations among different population groups, indicate a mean value of vitamin D content in breast milk and/or “anti-rachitic activity” of 45 IU/L (range: 14–88 IU/L) [47,49,50]. In other words, the vitamin D content in breast milk is clearly too low to meet the established vitamin D requirement of 400 IU per day throughout the first year of life [17,39,51]. Moreover, it has recently been noted that the vitamin D content in breast milk, both cholecalciferol and 25(OH)D, seems to have been progressively decreasing in recent decades in relation to changes in lifestyle: less exposure to solar radiation because of health and/or sociocultural reasons [48,50].

Although it is known that exposure to solar radiation can increase the vitamin D content in breast milk, there are few references on the effects of long-term continuous sun exposure on vitamin D content in breast milk, given the risk of skin carcinogenesis associated with continued exposure to solar ultraviolet radiation. In fact, research has focused on whether maternal vitamin D supplementation could not only improve the vitamin D status of the mother but also increase the vitamin D content in breast milk to a level that could supply infants with their age-standardized vitamin D requirements. Several randomized controlled clinical trials have verified that supplementation of lactating mothers with high doses of vitamin D (6000–6400 IU daily) significantly increases the antirachitic activity of breast milk, reaching values of vitamin D content in breast milk of more than 800 IU/L [34,52]. The US Endocrine Society currently recommends a daily dose of 1500–2000 IU for breastfeeding mothers to meet their own needs but warns that if it is not possible for the infant to be supplemented with vitamin D (400 IU daily), the mothers of these infants should be supplemented with 4000–6000 IU daily of vitamin D to meet the needs of their children [17].

## 10. Pharmacological Vitamin D Supplementation in the Infant

Even if circulating levels of 25(OH)D in pregnant women were sufficient, vitamin D stores in the newborn are practically depleted by 6–8 weeks of postnatal life; thus, sun exposure and breast milk would constitute the natural sources for an infant during the first months of life. It is currently recommended that infants under six months of age should not be exposed to direct sunlight as the most appropriate photoprotection measure to reduce the risks of skin cancer [53]; therefore, they need pharmacological vitamin D supplementation. In fact, in exclusively breastfed infants without vitamin D supplementation, the prevalence of vitamin D deficiency ranged from 67% in Japan to 76% in Ohio (United States) and 82% in the United Arab Emirates [49,52].

In 2010, the IOM, after relevant randomized clinical trials [54], updated its previous recommendations (200 IU daily) and advised increasing the prophylactic dose of vitamin D (cholecalciferol) to 400 IU daily. These recommendations have been adopted by the American Academy of Pediatrics [55], the US Endocrine Society [17], the ESPGHAN (The European Society for Paediatric Gastroenterology Hepatology and Nutrition) [56] and, in our country (Spain), the Spanish Association of Pediatrics [57]. In other words, it is currently recommended that newborns should start taking a pharmacological oral cholecalciferol supplement of 400 IU daily as soon as possible, which should be maintained during the first year of life to ensure an adequate supply of vitamin D, the effectiveness of which has been accredited by different authors [51,58,59]. However, it has also been described that the administration of an oral megadose of 50,000 IU of vitamin D in the newborn (Table 1) achieves similar results to supplementation with 400 IU daily of vitamin D in infants without evidence of adverse effects [60,61].

Other alternatives have been tested, such as replacing pharmacological supplementation of the infant with vitamin D with exclusively maternal supplementation, since improving maternal vitamin D status would simultaneously increase the antirachitic activity of breast milk (Table 1). In fact, several randomized controlled clinical trials have verified that maternal supplementation of high oral doses of vitamin D (6400 IU daily) not only significantly increases maternal circulating levels of 25(OH)D and the antirachitic activity of breast milk but also achieves levels of 25(OH)D in the infant similar to those achieved with supplementation of 400 IU daily of vitamin D in the infant, and without evidence of adverse effects [34,62,63]. Several randomized clinical trials have also evaluated the effects of exclusively maternal and intermittent supplementation of oral vitamin D megadose. For example, with a dose of 60,000 IU of vitamin D in the immediate postpartum period and at 6, 10, and 14 weeks after delivery (240,000 IU in total) [19] or with a monthly dose of 120,000 IU of vitamin D [64], or with a daily dose of 60,000 IU during the 10 days after delivery (600,000 IU in total) [65], with similar results to supplementation with 400 IU daily of vitamin D in infants, and with no adverse effects reported.

**Table 1 ijms-24-11881-t001:** Alternatives to recommended cholecalciferol supplementation (400 IU/day) in exclusively breastfed infants.

Exclusive Supplementation in the Infant	
Huynh et al., 2017 [61]	50,000 UI in the newborn (single dose)
Mother exclusive supplementation	
Hollis et al., 2015 [34]	6500 UI daily
Chandy et al., 2016 [64]	120,000 UI monthly
Naik et al., 2017 [65]	60,000 UI daily for the first 10 days after delivery (total of 600,000 UI)
Trivedi et al., 2020 [24]	60,000 UI in immediate postpartum and after 6, 10 and 14 weeks (total of 240,000 UI)

However, there is still some concern about supplementation with high-dose and/or megadose of maternal vitamin D, despite the lack of adverse effects, perhaps because these doses are above the established upper limit of 4000 IU/day for lactating mothers [39]; however, these strategies have failed to gain wide acceptance in clinical practice [66]. Although the risk of vitamin D intoxication is unlikely (intoxication would occur if the nursing mother ingested a vitamin D supplement exceeding 10,000 IU daily for a prolonged period), given its potential negative consequences (hypercalcemia, hypercalciuria, and hyperphosphatemia, which, in turn, are responsible for soft-tissue and vascular calcification and nephrolithiasis in the long term) it is important to be aware of this circumstance and, when appropriate, to prescribe the tolerable upper level of vitamin D always established under medical supervision [2]. Future clinical trials will be necessary to ensure the minimum effective dose for the prevention of vitamin D deficiency in the infant through maternal supplementation with high-dose vitamin D, as well as to establish the safety of such regimens for use in the general population. These alternatives could be especially useful in those situations in which, for various reasons (social, cultural, economic, etc.), adherence to daily pharmacological supplementation in the infant would be particularly difficult.

## 11. Conclusions

Although vitamin D deficiency is a global health problem, it requires special consideration in pregnant and lactating mothers because of its potential adverse maternal–fetal consequences. Vitamin D metabolism during gestation is exceptionally unique, manifesting apparent “physiological mismatches” that would correspond to maternal–fetal adaptive mechanisms of an epigenetic nature rather than playing a role in the regulation of calcium homeostasis.

Maternal serum 25(OH)D levels are directly related to fetal and/or neonatal 25(OH)D levels; vitamin D deficiency in the pregnant mother has been linked to various maternal–fetal pathologies (preeclampsia and gestational diabetes, prematurity, and/or intrauterine growth retardation, and poor skeletal development) and the risk of poor fetal and/or postnatal vitamin D status. In addition, vitamin D deficiency in utero could play an epigenetic influence and condition various pathologies in the long term.

Vitamin D content in breast milk is clearly insufficient to meet the vitamin D requirements of the infant; therefore, infants fed exclusively with breast milk are susceptible to vitamin D deficiency. For these reasons, most scientific societies recommend oral pharmacological supplementation with a daily dose of 400 IU of vitamin D (cholecalciferol) in infants during the first year of life. As an alternative, they recommend exclusive maternal and intermittent supplementation with high oral doses of vitamin D in order to increase the vitamin D content in breast milk and reduce the risk of hypovitaminosis D in the infant; however, these alternatives have not achieved great acceptance in clinical practice.

The explanation—at least theoretical—for the prevalence of vitamin D deficiency in pregnant women, as well as the low vitamin D content in breast milk with respect to the vitamin D needs of infants, is mainly related to the changes that have occurred in human life habits in recent decades (little outdoor activity, always wearing clothes, and often using sunscreen). This eventuality on an evolutionary scale would be an insignificant time for the human species, which would not provide the necessary time for biological adaptation to these changes in lifestyles and, consequently, it would be practically impossible for nursing mothers to synthesize sufficient vitamin D to ensure that the infant’s needs are met.

## Figures and Tables

**Figure 1 ijms-24-11881-f001:**
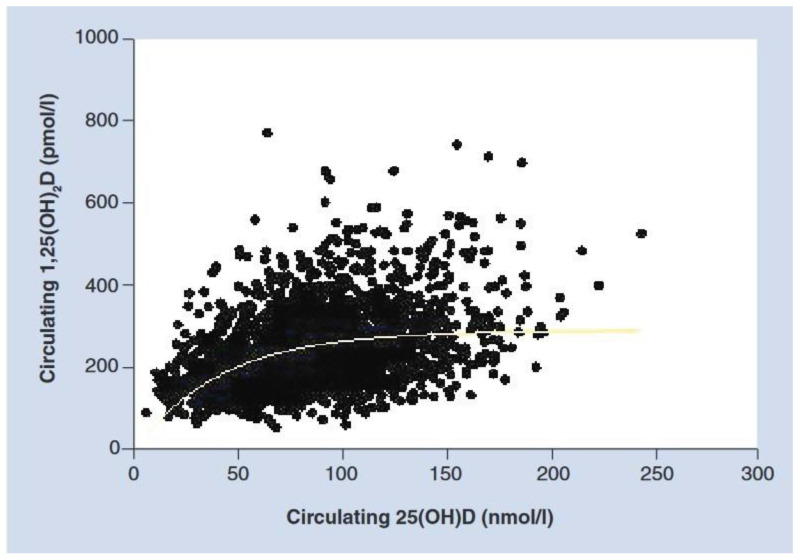
Relationship of circulating 25(OH)D on circulating 1,25(OH)2D during pregnancy (Hollis et al., 2011 [19] with permission).

## Data Availability

The datasets generated during and/or analyzed during the current study are available from the corresponding author upon reasonable request.

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
