# Peer review of "Pregnancy, Breastfeeding, and Vitamin D"

_ijms, 2023, doi:10.3390/ijms241511881_

Round 1
Reviewer 1 Report
Critique IJMS-2515647
Pregnancy, breastfeeding and vitamin D
Teodoro Dura-Tave and Fidel Gallinas-Victoriano
Note: From Special Issue on: The Role of Vitamin D in Human Health and Diseases
This relatively short review is part of a package of reviews on the important topic of vitamin D and its role in humans beyond to well-studied topic of bone growth and regression.
The area of vitamin D activity outside of bone biology remains controversial. This review by Dura-Tave and Gallinas-Victoriano, deals more with the metabolism of vitamin D both in the mother and in the fetus during pregnancy. I personally am not familiar with how vitamin D is handled during pregnancy and thus it provided important new information as I believe it will for many other biomedical researchers who study other aspects of vitamin D action.
What was disturbing was the documented finding that vitamin D content in human breast milk is low and insufficient to obtain the level of active vitamin D3 of 400 IU daily. Checking of other literature verifies the authors conclusion.
The only addition I would have liked to have seen is potential effects on the fetus, beyond bone problems, i.e. would the supplementation of vitamin B to achieve the 400IU daily also allow the normal action of vitamin D3 on immune system and in controlling inflammation.
Author Response
Comments and Suggestions for Authors
This relatively short review is part of a package of reviews on the important topic of vitamin D and its role in humans beyond to well-studied topic of bone growth and regression.
The area of vitamin D activity outside of bone biology remains controversial. This review by Dura-Tave and Gallinas-Victoriano, deals more with the metabolism of vitamin D both in the mother and in the fetus during pregnancy. I personally am not familiar with how vitamin D is handled during pregnancy and thus it provided important new information as I believe it will for many other biomedical researchers who study other aspects of vitamin D action.
What was disturbing was the documented finding that vitamin D content in human breast milk is low and insufficient to obtain the level of active vitamin D3 of 400 IU daily. Checking of other literature verifies the authors conclusion.
The only addition I would have liked to have seen is potential effects on the fetus, beyond bone problems, i.e. would the supplementation of vitamin B to achieve the 400 IU daily also allow the normal action of vitamin D3 on immune system and in controlling inflammation.
We would like to thank you for your encouraging words regarding this article.
With regard to your concern, we would like to share with you, as referred to in the original text. (1. Introduction):
“…at present day, vitamin D is considered a pleitropic hormone. In fact, in addition to the recognized role in calcium homeostasis and bone metabolism, vitamin D regulates the expression of a variety of genes that control multiple metabolic pathways related to immune function, as well as proliferation, differentiation and cell metabolism. This information helps explain the fundamentals of the large number of recent publications that associate vitamin D deficiency with a higher risk of several chronic diseases (autoimmune, cardiovascular, infectious, metabolic, neurologic, psychiatric diseases and several types of cancer) and therefore justify the concern on the monitoring of its organic content.”
Later, the original text refers: (3. Definition of hypovitaminosis D):
“…Even though it is generally accepted that serum 25(OH)D levels are the best indicator of body vitamin D stores, there is some controversy with regard to the limits that define normality. In 2011, the American Institute of Medicine (IOM) proposed as reasonable to consider 25(OH)D levels of 20 ng/ml as the threshold of normality of the organic vitamin D content, given the practical absence of significant metabolic alterations at concentrations above 20 ng/ml, while levels below these figures induce an increase in PTH and bone alterations might be already detected. However, in 2012 the US Endocrine Society considered it prudential, in order to safeguard the multiple extra-skeletal functions of vitamin D, to modify the 25(OH)D values defining vitamin D sufficiency, without modifying the already established cut-off point for vitamin D deficiency, but intercalating between these two states the concept of vitamin D insufficiency, which should also be overcome.
These new cut-off points, although based on observational studies, are being accepted and used by most authors:
- Vitamin D deficiency: calcidiol levels are lower than 20 ng/ml (<50 nmol/L).
- Vitamin D insufficiency: calcidiol levels range between 20 and 29 ng/ml (51-74 nmol/L).
- Vitamin D sufficiency: calcidiol levels are equal to or higher than 30 ng/ml (>75 nmol/L).
In a recent review (Palacios C, et al. Cochrane Database Sys Rev, 2019) cited in the present work (ref. 10) it is indicated that vitamin D supplementation, in order to reach levels above 30 ng / ml, reduces the risk of pre-eclampsia, gestational diabetes, low birthweight and may reduce the risk of severe postpartum hemorrhage. It means, this increase in serum 25-hydroxyvitamin D concentrations in pregnancy may explain the potential health benefits on maternal and neonatal health outcomes. However, it concludes that additional rigorous high quality and larger randomized trials are required to evaluate the effects of vitamin D supplementation in pregnancy (maternal and infant outcomes). In addition, trials are needed to evaluate systematically maternal adverse events to confirm the safety of the supplementation. Information on the most effective and safe dosage, the optimal dosing regimen (daily, intermittent or single doses), the timing of initiation of vitamin D supplementation, and the effect of vitamin D when combined with other vitamins and minerals are also needed to inform policy-making.
Finally, it should be noted that in the original text (10. Pharmacological supplementation of vitamin D in the infant) is discussed:
“…In 2010 the IOM, after relevant randomized clinical trials, updated its previous recommendations (200 IU daily) and advised increasing the prophylactic dose of vitamin D (cholecalciferol) to 400 IU daily. These recommendations have been adopted by the American Academy of Pediatrics (55), the US Endocrine Society, the ESPGHAN (The European Society for Paediatric Gastroenterology Hepatology and Nutrition) and, in our country (Spain), by the Spanish Association of Pediatrics. In other words, it is currently recommended that newborns should start taking a pharmacological oral cholecalciferol supplement of 400 IU daily as soon as possible, which should be maintained during the first year of life to ensure an adequate supply of vitamin D, the effectiveness of which has been accredited by different authors….”
Therefore, as a conclusion, it should be noted that it would be a question of achieving levels of calcidiol equal to greater than 30 ng / ml (vitamin D sufficiency), both in the pregnant/lactating woman and in the infant, to hypothetically safeguard its different extra-skeletal biological functions and that would obviously include its modulating function on the innate and adaptive immune systems and anti-inflammatory effects.
We would like to express our thanks to the referee for your suggestions and positive criticisms.
We hope every made question have been answered adequately.
Yours sincerely,
Teodoro Durá-Travé
Reviewer 2 Report
In this manuscript, guidelines for vitamin D supplementation and the impact of reduced sunlight exposure are discussed. The manuscript is well written with decent logic, though the authors need to address some concerns before it is considered for publication.
1. Throughout this entire manuscript, only one table is presented. It is recommended to incorporate additional figures in this review to enhance the visual representation of the content.
2. Typos: In line 29, "requires" should be corrected to "require."
3. The introduction section would benefit from the inclusion of additional references to enhance its scholarly foundation and provide a comprehensive overview of the topic.
Quality of English is decent.
Author Response
2-Comments and Suggestions for Authors
In this manuscript, guidelines for vitamin D supplementation and the impact of reduced sunlight exposure are discussed. The manuscript is well written with decent logic, though the authors need to address some concerns before it is considered for publication.
- Throughout this entire manuscript, only one table is presented. It is recommended to incorporate additional figures in this review to enhance the visual representation of the content.
One figure (figure 1) has been added to enhance the visual representation of the content.
Figure 1. Relationship of circulating 25(OH)D on circulating 1,25(OH)2D during pregnancy (Hollis et al., 2011 [19] with permission).
In line 29, "requires" has been corrected to "require."
- The introduction section would benefit from the inclusion of additional references to enhance its scholarly foundation and provide a comprehensive overview of the topic.
Additional references have been added in the Introduction section of the second version to provide a comprehensive overview of the topic.
The following paragraph has been added:
Vitamin D metabolism manifests significant changes in pregnant women in comparison to the non-pregnant state (2, 9, 12, 13), but several questions about the role of vitamin D in pregnancy remain unanswered. Vitamin D deficiency has been reported among pregnant women and nursing mothers globally, constituting a risk group for vitamin D deficiency (2, 18, 25, 26). Vitamin D deficiency during pregnancy has been associated not only with pregnancy outcomes, but also with posterior physical and mental health of the offspring (17, 30, 31, 32).
We would like to express our thanks to the referees for your suggestions and positive criticisms.
We hope every made question have been answered adequately.
Yours sincerely,
Teodoro Durá-Travé
Reviewer 3 Report
Good article and well written, however, for completeness it would be appropriate to also indicate a chapter on the risks of excess Vitamin D supplementation
Author Response
Comments and Suggestions for Authors
Good article and well written, however, for completeness it would be appropriate to also indicate a chapter on the risks of excess Vitamin D supplementation
First, we would like to thank you for your encouraging words regarding this article.
After carefully analyzing the reviewer's suggestion, we understand that it would not be necessary, given the objectives of this review article, to add a specific chapter on the toxicity of vitamin D. However, a paragraph has been included that expressly refers to this suggestion of the reviewer.
Previous sentences (last paragraph of the chapter “Pharmacological supplementation of vitamin D in the infant”) …
However, there is still some concern about supplementation with high-dose and/or megadose of maternal vitamin D, despite the lack of adverse effects, perhaps because these doses are above the established upper limit of 4,000 IU/day for lactating mothers (39); but, in fact, these strategies have failed to gain wide acceptance in clinical practice (66). Future clinical trials will be necessary to ensure the minimum effective dose for the prevention of vitamin D deficiency in the infant through maternal supplementation with high-dose vitamin D, as well as to establish the safety of such regimens for use in the general population. These alternatives could be especially useful in those situations in which for various reasons (social, cultural, economic, etc.) adherence to daily pharmacological supplementation in the infant would be particularly difficult.
…have now been changed into:
However, there is still some concern about supplementation with high-dose and/or megadose of maternal vitamin D, despite the lack of adverse effects, perhaps because these doses are above the established upper limit of 4,000 IU/day for lactating mothers (39); but, in fact, these strategies have failed to gain wide acceptance in clinical practice (66). Although the risk of vitamin D intoxication is unlikely (intoxication would occur if the nursing mother ingested a vitamin D supplement exceeding 10,000 IU daily for a prolonged period), given its potential negative consequences (hypercalcemia, hypercalciuria, and hyperphosphatemia, which, in turn, are responsible for soft-tissue and vascular calcification and nephrolithiasis in the long term) it is important to be aware of this circumstance and, when appropriate, to prescribe the tolerable upper level of vitamin D established always under medical supervision. Future clinical trials will be necessary to ensure the minimum effective dose for the prevention of vitamin D deficiency in the infant through maternal supplementation with high-dose vitamin D, as well as to establish the safety of such regimens for use in the general population. These alternatives could be especially useful in those situations in which for various reasons (social, cultural, economic, etc.) adherence to daily pharmacological supplementation in the infant would be particularly difficult.
We would like to express our thanks to referee for your suggestions and positive criticisms.
We hope every made question have been answered adequately.
Yours sincerely,
Teodoro Durá-Travé